# Metal/Semiconductor Nanocomposites for Photocatalysis: Fundamentals, Structures, Applications and Properties

**DOI:** 10.3390/nano9030359

**Published:** 2019-03-04

**Authors:** Yong-sheng Fu, Jun Li, Jianguo Li

**Affiliations:** School of Materials Science and Engineering, Shanghai Jiao Tong University, Shanghai 200240, China; fuys648@163.com (Y.-s.F.); lijg@sjtu.edu.cn (J.L.)

**Keywords:** nanocomposite photocatalyst, environmental remediation, selective organic transformation, hydrogen evolution, disinfection

## Abstract

Due to the capability of utilizing light energy to drive chemical reactions, photocatalysis has been widely accepted as a green technology to help us address the increasingly severe environment and energy issues facing human society. To date, a large amount of research has been devoted to enhancing the properties of photocatalysts. As reported, coupling semiconductors with metals is one of the most effective methods to achieve high-performance photocatalysts. The excellent properties of metal/semiconductor (M/S) nanocomposite photocatalysts originate in two aspects: (a) improved charge separation at the metal-semiconductor interface; and (b) increased absorption of visible light due to the surface plasmon resonance of metals. So far, many M/S nanocomposite photocatalysts with different structures have been developed for the application in environmental remediation, selective organic transformation, hydrogen evolution, and disinfection. Herein, we will give a review on the M/S nanocomposite photocatalysts, regarding their fundamentals, structures (as well as their typical synthetic approaches), applications and properties. Finally, we will also present our perspective on the future development of M/S nanocomposite photocatalysts.

## 1. Introduction

Since the discovery of water splitting on a TiO_2_ electrode under irradiation of ultraviolet (UV) light by Fujishima and Honda in 1972, many semiconductors, such as CdS, ZnO, SrTiO_3_ and g-C_3_N_4_, have been demonstrated to exhibit photocatalytic properties and, through the efforts of researchers, the application areas of semiconductor photocatalysts have been greatly extended [1,2,3,4,5,6,7,8,9,10,11,12,13,14]. Unfortunately, due to fast charge separation and limited light absorption, the properties of semiconductor photocatalysts are relatively unsatisfactory for practical application [15,16,17]. To improve the properties of semiconductor photocatalysts, several strategies have been proposed by researchers, such as doping, dye-sensitization, coupling, etc. [18,19,20,21,22,23,24,25,26,27]. However, each of these strategies has its own pros and cons. Toward the strategy of doping, the band structures of semiconductor photocatalysts could be modulated by the doping atoms to exhibit better properties for light absorption, but the doped semiconductor photocatalysts could be more susceptible to photo-corrosion and the charge recombination of the doped semiconductor photocatalysts could be aggravated at the defects introduced by the doping atoms [19,21,27]. As for the strategy of dye-sensitization, although the light-sensitive dyes can absorb light more efficiently and transfer the photoexcited electrons to the semiconductor photocatalysts, the light-sensitive dyes are susceptible to chemical corrosion, resulting in the poor stability of dye-sensitized semiconductor photocatalysts [24,25,26]. Coupling, as a strategy with a relatively short history, has aroused great interest among researchers since its very beginning. When coupled with metals, especially the noble metals, the properties of semiconductor photocatalysts can be considerably improved due to the enhanced charge separation at the metal–semiconductor interface and the enhanced visible light absorption caused by the surface plasmon resonance (SPR) of metals [28,29,30]. Except for the high cost of noble metals, the main drawback of the coupling strategy used to be the poor control of the process of coupling semiconductors with metals. However, with the development of nanotechnology, the coupling process now could be delicately controlled and several new structures have been synthesized to improve the properties of metal/semiconductor (M/S) nanocomposite photocatalysts [29,31].

Herein, we would like to review the M/S nanocomposite photocatalysts regarding their fundamentals, structures (as well as their typical synthetic approaches), applications and properties. Finally, we will also present our perspective on the future development of M/S nanocomposite photocatalysts.

## 2. Fundamentals

### 2.1. Principles of Photocatalysis

As defined by the International Union of Pure and Applied Chemistry (IUPAC), photocatalysis is “A catalytic reaction involving light absorption by substrate” [32]. When semiconductor photocatalysts are irradiated by light with the photon energy larger than their band gap (BG) energy, the electrons in the valence band (VB) will be excited into the conduction band (CB), leaving positive holes in the VB. Because photocatalytic reactions are reactions happening at the surface of photocatalysts, the photo-induced free charge carriers need first to diffuse into the active sites on the surface of photocatalysts before they can induce photocatalytic reactions [33,34,35]. The photocatalytic process is schematically presented in Figure 1. However, for a particular substrate, whether it can undergo chemical reactions on the semiconductor photocatalysts depends on the relative positions between its redox potentials and the band edges of semiconductor photocatalysts [36,37,38]. There are four possibilities as follows:(1)If the redox potential of the substrate is lower than the CB edge of the semiconductor photocatalyst, then the substrate can undergo reductive reactions.(2)If the redox potential of the substrate is higher than the VB edge of the semiconductor photocatalyst, then the substrate can undergo oxidative reactions.(3)If the redox potential of the substrate is higher than the CB edge or lower than the VB of the semiconductor photocatalyst, then the substrate can undergo neither reductive nor oxidative reactions.(4)If the redox potential of the substrate is lower than the CB edge and higher than the VB of the semiconductor photocatalyst, then the substrate can undergo either reductive or oxidative reactions.

### 2.2. Mechanisms for the Enhanced Properties of Metal/Semiconductor (M/S) Nanocomposite Photocatalysts

As mentioned above, by coupling with different metals, the mechanisms for the enhanced properties of M/S nanocomposite photocatalysts can be divided into two categories, i.e., the enhanced charge separation at the metal-semiconductor interface and the enhanced visible light absorption due to the SPR of metals.

#### 2.2.1. Enhanced Charge Separation

The enhanced charge separation of M/S nanocomposite photocatalysts originates from the electron transfer across the metal–semiconductor interface [31,39,40,41]. When a semiconductor is coupled with a metal, which possesses a higher work function than that of the semiconductor (it is the most common case for M/S nanocomposite photocatalysts), then the electrons will flow from the semiconductor to the metal until their Fermi levels (i.e., E_F,m_ and E_F,s_) are aligned, leading to the upward bending of the band edges in the semiconductor as revealed in Figure 2. Accompanied with the band bending, a Schottky barrier (i.e., ϕ_SB_ in Figure 2) forms at the metal–semiconductor interface [39]. As has been shown, the efficiency of electron transfer across the metal–semiconductor interface increases with the height of the Schottky barrier [42]. In addition, the metals in M/S nanocomposite photocatalysts not only act as the electron reservoir to enhance the charge separation of the semiconductors, but also provide active sites for reductive reactions, hence dramatically improving the properties of M/S nanocomposite photocatalysts [43].

#### 2.2.2. Enhanced Visible Light Absorption

When coupled with metals such as Ag and Au, the SPR of metals could endow the M/S nanocomposite photocatalysts with enhanced absorption toward visible light [44,45]. However, the mechanism governing the SPR-enhanced visible light absorption of M/S nanocomposite photocatalysts is still unclear [28,29,46,47]. So far, there exist three non-mutually exclusive mechanisms for the explanation of the SPR-enhanced properties of M/S nanocomposite photocatalysts under visible light irradiation, i.e., (a) SPR-induced electron injection from metals to semiconductors; (b) charge separation induced by near-field electric field (NFEF); (c) scattering-enhanced light absorption.

##### Surface Plasmon Resonance (SPR)-Induced Electron Injection

Due to the SPR of metals such as Au and Ag, the energy of excited electrons in the metals can be excited to the range between 1.0 and 4.0 eV above their Fermi levels. Then, the energetic electrons will overcome the Schottky barrier at the metal–semiconductor interface and transfer to the CB of semiconductors, leaving energetic holes in the metals, as schematically shown in Figure 3a. Thus, the hot electrons in the semiconductors will drive reduction reactions and the hot holes in the metals will drive oxidation reactions.

##### Charge Separation Induced by Near-Field Electric Field (NFEF)

Under visible light irradiation, the SPR-excited hot electrons in the metal particles of M/S nanocomposite photocatalysts will generate intense electric field in their proximity (i.e., NFEF), which could be up to 100–10000 times larger than the electric field of incident light. Since the formation rate of electron-hole pairs in the semiconductors of M/S nanocomposite photocatalysts is proportional to the intensity of the NFEF (more specifically, |E|^2^), the generation of electron-hole pairs in M/S nanocomposite photocatalysts can be significantly enhanced. The process of NFEF-induced charge separation is schematically shown in Figure 3b.

##### Scattering-Enhanced Light Absorption

If the size of the metal particles in M/S nanocomposite photocatalysts is larger than 50 nm, the SPR-excited metal particles can efficiently scatter the incident light, leading to the increase of the path length of light through the M/S nanocomposite photocatalysts. Thus, the light absorption of M/S nanocomposite photocatalysts will be improved due to the increased path length of light, resulting in the enhancement of the properties of M/S nanocomposite photocatalysts. Figure 3c schematically reveals the scattering-enhanced photon absorption mechanism.

## 3. Structures of M/S Nanocomposite Photocatalysts

Based on the morphologies and synthetic approaches, the structures of M/S nanocomposite photocatalysts could be divided into six categories, i.e., the conventional structure, core-shell structure, yolk-shell structure, Janus structure, array structure and multi-junction structure, which are schematically shown in Figure 4.

### 3.1. Conventional Structure

The conventional structure of M/S nanocomposite photocatalysts refers to the structure, in which the combination of semiconductor nanoparticles and metal nanoparticles is not delicately controlled. So far, many approaches have been developed to synthesize M/S nanocomposite photocatalysts with conventional structure, which include photoreduction, impregnation, deposition-precipitation, chemical vapor deposition (CVD) etc. [48,49,50,51].

#### 3.1.1. Photoreduction

Since some chemical compounds containing metal element can be reduced to metal under irradiation, photoreduction is regarded as a facile approach to decorate semiconductor nanostructures with noble metal. Typically, in the work of Yu et al., hydrothermally-synthesized TiO_2_ nanosheets were added to an aqueous solution of H_2_PtCl_6_. Under stirring and the irradiation of ultraviolet (UV) light, H_2_PtCl_6_ was reduced to Pt and deposited on TiO_2_ nanosheets, leading to the formation of Pt-decorated TiO_2_ nanosheets [48].

#### 3.1.2. Impregnation

During the impregnation process, the combination of noble metal and semiconductor nanostructures is achieved through the attachment of noble metal precursor onto semiconductor substrate due to the van der Waals force, coulomb force or some other interactions between them. Typically, in the work of Lin et al., H_2_PtCl_6_ was added into a suspension of TiO_2_ (B) nanofibers. Then the solvent was evaporated under stirring. Then, the as-obtained powders were heated in a reducing atmosphere to reduce H_2_PtCl_6_ absorbed on TiO_2_ (B) nanofibers, leading to the formation of Pt-decorated TiO_2_ (B) nanofibers [49].

#### 3.1.3. Deposition-Precipitation

Deposition-precipitation is also a facile approach to synthesize M/S nanocomposites. Typically, in the work of Wu et al., TiO_2_ nanoparticles were dispersed in an aqueous solution of chloroauric acid. Then NaOH solution was added into the suspension to adjust the pH value to the desired level. After stirring, the precipitates were filtered, washed and dried. Then, the as-obtained products were calcined to form the Au-decorated TiO_2_ nanoparticles [50].

#### 3.1.4. Chemical Vapor Deposition (CVD)

As is well known, CVD is a powerful approach in synthesizing nanostructure and is often utilized to prepare metal-decorated nanostructures. Typically, in the work of Shi et al., HAuCl_4_ was vaporized at a high temperature in a tube furnace. Then, the vapor was blown at the TiO_2_ nanorod arrays at a relatively low temperature with the carrier gas of nitrogen. Thus, the gold would deposit on the TiO_2_ nanorods, leading to the formation of Au-decorated TiO_2_ nanorod arrays [51].

### 3.2. Core-Shell Structure

The core-shell structure was first developed to improve the quantum yield of quantum dots in the 1990s [52]. After that, the core-shell structure attracted great interest from researchers and its application areas were greatly extended. In photocatalysis, M/S nanocomposite photocatalysts with core-shell structure occupy an important position due to their outstanding photocatalytic properties [53]. Besides, the enhanced charge transfer between metal and semiconductor, the core-shell structure can also hinder the aggregation of particles and protect the metal core from undesired corrosion or dissolution during the photocatalytic process.

The synthesis of core-shell M/S nanoparticles usually involves the coating of a semiconductor layer on metal nanoparticles. For example, in the work of Sudeep, photocatalytic Ag@TiO_2_ core-shell nanoparticles were synthesized through controlled hydrolysis of titanium-(triethanolaminato) isopropoxide (TTEAIP) on the surface of Ag nanoparticles. Briefly, TTEAIP and silver nitrate were added into 2-propanol followed by stirring. Then dimethyl formamide (DMF) was added to the solution. Next, the solution was heated and refluxed. During this process, silver nitrate was first reduced to Ag nanoparticles by DMF. Then, due to the interaction between Ag nanoparticles and triethanolamine ligands, TTEAIP hydrolyzed on the surface of Ag nanoparticles, leading to the formation of Ag@TiO_2_ core-shell nanoparticles [54].

### 3.3. Yolk-Shell Structure

The yolk-shell structure originates from the pioneer work of Xia’s group as a variation of the core-shell structure [55]. Under the efforts of researchers, the yolk-shell structure has shown promising application in nanoreactors, drug delivery and lithium ion batteries [56]. Recently, some researchers have also investigated the application of yolk-shell structures in photocatalysis [57,58,59].

The synthesis of yolk-shell structures usually needs a sacrificial template. In the work of Li et al., gold nanoparticles were first coated with a layer of SiO_2_ by the hydrolysis of tetraethyl orthosilicate (TEOS). Next, the SiO_2_-coated gold nanoparticles were coated with a layer of TiO_2_ by the hydrolysis of tetrabutyl titanate (TBOT). Then, the as-prepared gold nanoparticles were coated with a layer of SiO_2_ again to protect the TiO_2_ layer during the calcination process. After calcination, the two SiO_2_ layers were removed by the etch of NaOH solution. Thus, the Au@TiO_2_ yolk-shell nanoparticles were prepared. The SiO_2_ layers served as the sacrificial template during the synthetic process [57].

### 3.4. Janus Structure

Janus particles, as first described by de Gennes in 1991, refer to the particles with anisotropic structure, which are composed of two distinct parts [60]. The dual nature of Janus particles endows themselves with fascinating properties, such as unique surface properties, controlled self-assembly behavior and response to multiple stimuli, etc. [61,62]. When applied in photocatalysis, Janus nanoparticles composed of metal and semiconductor can also exhibit remarkable photocatalytic properties.

Different from conventional structures, the synthesis of Janus structures needs delicate control over the combination of two distinct parts. As demonstrated by the work of Seh et al., Au/TiO_2_ Janus nanoparticles were synthesized through controlled hydrolysis of titanium diisopropoxide bis(acetylacetonate) (TAA) on the surface of gold nanoparticles. The reason for choosing TAA was that the hydrolysis rate of TAA was rather slow, which had a significant influence on the structure of the products. Thus, during the slow hydrolysis of TAA in the alkaline suspension of gold nanoparticles in isopropanol, TiO_2_ combined with gold nanoparticles to form the Au/TiO_2_ Janus nanoparticles [63].

### 3.5. Array Structure

Due to the promising application in nanodevices, the array structure has been widely researched during the past few decades [64,65,66]. In photocatalysis, the array structure has also been utilized to improve the photocatalytic properties of M/S nanocomposites [67].

The synthesis of array structures is mainly achieved through a template-assisted deposition process. In the work of Wang et al., an anodic aluminum oxide (AAO) template with one side deposited with a layer of gold and connected to a piece of aluminum foil was utilized as the working electrode in an electrochemical cell. For the deposition of CdS, the working electrode was biased to −2.5 V vs. standard calomel electrode (SCE) in the electrolyte of sulfur and cadmium chloride dissolved in dimethyl sulfoxide (DMSO). For the deposition of Au, the working electrode was biased to –0.95 V vs. SCE in commercial Au Orotemp 24. After alternate deposition of CdS and Au, the AAO template was removed through chemical etching. Thus, photocatalytic Au/CdS nanorod arrays were obtained [67].

### 3.6. Multi-Junction Structure

M/S nanocomposite photocatalysts with multi-junction structures are formed by sandwiching metal nanoparticles between two semiconductors. Thus, the electron transfer process in multi-junction M/S nanocomposite photocatalysts is very similar to that of Z-scheme photocatalytic systems in nature, which could endow them excellent photocatalytic properties [68,69,70].

Take the pioneer work of Tada et al. for example. The CdS/Au/TiO_2_ multi-junction nanoparticles as the all-solid-state Z-scheme photocatalytic system were synthesized through a two-step deposition process. In the first step, gold was deposited on the TiO_2_ nanoparticles through an impregnation process with chloroauric acid as the precursor. In the second step, cadmium sulfide was deposited on the Au/TiO_2_ nanoparticles through the UV-induced reaction of sulfur and cadmium perchlorate. Thus, CdS/Au/TiO_2_ multi-junction nanoparticles were prepared [71].

## 4. Applications and Properties of M/S Nanocomposite Photocatalysts

To date, the application areas of photocatalysis are mainly focused on environmental remediation, selective organic transformation, hydrogen evolution and disinfection.

### 4.1. Environmental Remediation

The photo-excited electrons and holes in semiconductor photocatalysts as well as the associated radicals possess high chemical activity, which can degrade pollutants to low- or non-hazardous substances [72,73,74,75]. As reported, most of the researches in photocatalytic environmental remediation were focused on degradation of organic pollutants such as Rhodamine B (RhB), Methylene blue (MB), methyl orange (MO) and so on. To the best of our knowledge, the only inorganic pollutant that has been involved in the research of photocatalytic environmental remediation is nitric oxide (NO). Table 1 lists some of the most representative M/S nanocomposite photocatalysts applied in environmental remediation [51,59,71,76,77,78,79,80,81,82,83,84,85,86,87,88,89,90,91,92,93,94,95,96,97,98,99,100].

Due to the facile synthesis, more than half of the M/S nanocomposite photocatalysts listed in Table 1 possess the conventional structure. In the work of Lin et al., through coupling ZnO nanofibers with Ag, the as-synthesized Ag/ZnO photocatalysts exhibited much better properties for photocatalytic degradation of RhB than pristine ZnO photocatalysts, which was revealed in Figure 5a [79]. As revealed in Figure 5b, the properties of Ag/ZnO photocatalysts did not increase linearly with the Ag loading content and reached a peak at the Ag loading content of 7.5 at%. In addition, the repeatability test results in Figure 5c manifested the attenuation of the properties of Ag/ZnO photocatalysts was negligible after three cycles. The excellent properties of Ag/ZnO photocatalysts originated from the enhanced charge separation at the Ag-ZnO interface, which was evidenced by the photoluminescence (PL) spectra in Figure 5d. Toward removal of NO, Li et al. synthesized Bi/A-BO photocatalysts which could remove NO effectively under visible light illumination [86]. The remarkable properties of Bi/A-BO photocatalysts originated from SPR of Bi which could absorb visible light effectively and enhance the charge separation in A-BO. Li et al. also utilized electron spin resonance (ESR) spectroscopy for in-situ investigation of the reactive species during the photocatalytic process. (For detailed information about utilizing ESR spectroscopy to detect radicals, these two reviews, [101,102], are recommended.) The results revealed superoxide radicals (O_2_^−•^) radicals were the major active species for photocatalytic NO oxidation.

For the M/S nanocomposite photocatalysts possessing the core-shell structure, their properties in photocatalytic environmental remediation were in strong relationship with the metal core and shell thickness. In the work of Zhang et al., the effect of Au, Pd and Pt cores on the photocatalytic activity of M@TiO_2_ (M = Au, Pd and Pt) core-shell nanoparticles for degradation of RhB was investigated [91]. As revealed in Figure 6, core-shell M@TiO_2_ (M = Au, Pd and Pt) exhibited different photocatalytic properties under the irradiation of UV light and visible light. Under irradiation of UV light, the photocatalytic properties of core-shell M@TiO_2_ (M = Au, Pd and Pt) followed the order P25 TiO_2_ > Pt@TiO_2_ > Pd@TiO_2_ > Au@TiO_2_, while under irradiation of visible light the photocatalytic properties of core-shell M@TiO_2_ (M = Au, Pd and Pt) followed the order Pd@TiO_2_ > Pt@TiO_2_ > Au@TiO_2_ > P25 TiO_2_. The results indicated the metal core had two impacts on the photocatalytic properties of core-shell M@TiO_2_ (M = Au, Pd and Pt), i.e., improving charge separation at the metal-TiO_2_ interface and modulating light absorption of TiO_2_. In addition, Zhang et al. also demonstrated that the hydroxyl radicals (^•^OH) generated on core-shell M@TiO_2_ played a more important role than superoxide radicals and holes in degradation of RhB under irradiation of UV light, while under the irradiation of visible light holes generated on core-shell M@TiO_2_ played a predominant role in degradation of RhB. The effect of shell thickness on core-shell M/S nanocomposite photocatalysts was revealed by Kong et al. [93]. In their research, the properties of core-shell Au@Cu_2_O photocatalysts for degradation of MO varied with the Cu_2_O shell thickness in the order 127 nm > 96 nm > 197 nm > 250 nm > 158 nm. Kong et al. thought the nonlinear relationship between the properties of core-shell Au@Cu_2_O photocatalysts and the shell thickness could be attributed to several factors, i.e., surface area, light absorption and scattering, charge-carrier dynamics and core-shell interactions, which competed with each other.

Compared with M/S nanocomposite photocatalysts possessing the conventional and core-shell structures, there is less research on M/S nanocomposite photocatalysts with the yolk-shell, Janus and multi-junction structures for environmental remediation. Wang et al. investigated the properties of yolk-shell Au@TiO_2_ photocatalysts for degradation of RhB [59]. In their research, they also introduced reduced graphene oxide (r-GO) into the TiO_2_ shell. The results demonstrated the properties of yolk-shell Au@r-GO/TiO_2_ photocatalysts were better than yolk-shell Au@TiO_2_ photocatalysts, which indicated the charge separation in the TiO_2_ shell increased with the electron conductivity of the TiO_2_ shell. In the work of Yao et al., the properties of Janus Au/ZnO photocatalysts were investigated for degradation of MO [100]. The results demonstrated the properties of Janus Au/ZnO photocatalysts for degradation of MO were higher than that of pristine ZnO photocatalysts and increased with the size of Au nanoparticles under UV light illumination. In the pioneer work of Tada et al., multi-junction CdS/Au/TiO_2_ photocatalysts were synthesized as the all-solid-state Z-scheme photocatalysts for the degradation of MV [71]. As revealed in Figure 7a, multi-junction CdS/Au/TiO_2_ photocatalysts exhibited much higher properties than CdS/TiO_2_, Au/TiO_2_ and TiO_2_ photocatalysts, which revealed the charge separation caused by TiO_2_ → Au → CdS Z-scheme electron transfer was more efficient than that of the single- or double-component systems. The Z-scheme electron transfer was illustrated in Figure 7b. Besides, Tada et al. pointed out that the properties of all-solid-state Z-scheme photocatalysts could be further improved by modifying the energy band structures of the semiconductor components.

### 4.2. Selective Organic Transformation

Unlike the non-selective mineralization process in environmental remediation, photocatalysis can also drive organic transformation processes to selectively synthesize valuable chemicals [103,104,105]. Compared with thermochemical synthetic process, the photocatalytic selective organic transformation process often requires milder conditions and shorter reaction sequences and can exclude some undesirable side reactions. Based on the reported research, the photocatalytic selective organic transformation reactions can be divided into three categories, i.e., the oxidation reaction, reduction reaction and coupling reaction [103]. In Table 2, some of the most representative M/S nanocomposite photocatalysts applied in selective organic transformation are listed [106,107,108,109,110,111,112,113,114,115,116,117,118,119,120].

Towards the photocatalytic selective oxidation reactions, Yuzawa et al. investigated the effect of Pt loading on the properties of Pt/TiO_2_ photocatalysts for amination of benzene to aniline and unfolded the reaction mechanism governing the photocatalytic amination process [108]. As revealed in Figure 8, the conversion yield of Pt/TiO_2_ did not increase linearly with the Pt loading content and reached a peak at about 0.1 wt% Pt loading content, while the aniline selectivity of Pt/TiO_2_ photocatalysts was hardly influenced by the Pt loading content and remained about 97%. The mechanism governing the photocatalytic amination process was clarified through ESR spectroscopy. First, the holes on the TiO_2_ surface oxidized an ammonia molecule to form a neutral amide radical, which then reacted with the aromatic ring to form an intermediate, and afterwards the hydrogen of the intermediate was abstracted by the active sites on the Pt surface, leading to the formation of aniline. Thus, it could be deduced that the charge separation at the Pt-TiO_2_ interface could significantly enhance the properties of Pt/TiO_2_ photocatalysts for the amination of benzene. Zhang et al. investigated the properties of core-shell and yolk-shell Pt@CeO_2_ photocatalysts for oxidation of benzyl alcohol to benzaldehyde [111]. As revealed in Figure 9, core-shell Pt@CeO_2_ photocatalysts exhibited much higher benzaldehyde yield and selectivity than yolk-shell Pt@CeO_2_ photocatalysts. Zhang et al. attributed the unsatisfactory properties of yolk-shell Pt@CeO_2_ photocatalysts to the loose contact between the Pt core and CeO_2_ shell, which deteriorated the charge separation at the Pt-CeO_2_ interface.

As listed in Table 2, the photocatalytic selective reduction reactions mainly involve hydrogenation of nitroaromatics and reduction of CO_2_. The investigation of Tada et al. on hydrogenation of nitroaromatics demonstrated that the life time of electrons could also be prolonged by the charge separation at the Ag-TiO_2_ interface, leading to the enhanced properties of Ag/TiO_2_ photocatalysts for hydrogenation of nitrobenzene to aniline [113]. For better properties of photocatalytic reduction of CO_2_ to CH_4_, Xie et al. combined P25 TiO_2_ with five noble metals, i.e., Pt, Pd, Rh, Ag and Au [114]. During the photocatalytic test, the properties of M/TiO_2_ photocatalysts increased in the sequence of Ag/TiO_2_ < Rh/TiO_2_ < Au/TiO_2_ < Pd/TiO_2_ < Pt/TiO_2_, which indicated the charge separation at the metal-TiO_2_ interface improved with the Schottky barrier height.

Towards the photocatalytic coupling reactions, Jiao et al. investigated the properties of Pd/SiC photocatalysts for Suzuki coupling of iodobenzene and phenylboronic acid [120]. Under visible light illumination, the conversion of iodobenzene and selectivity for the main product of Pd/SiC both reached nearly 100%. Jiao et al. also evaluated the effect of photogenerated electrons and holes in Pd/SiC on the Suzuki coupling process. After adding the electron-capturing agent (or hole-capturing agent) into the reaction system, the photocatalytic properties of Pd/SiC decreased dramatically, which indicated both the photogenerated electrons and holes contributed to the Suzuki coupling process.

### 4.3. Hydrogen Evolution

As a significant process to convert solar energy into chemical energy, photocatalytic hydrogen evolution has always been the hot spot in the research field of photocatalysis [47,121,122,123]. Table 3 lists some of the most representative M/S photocatalysts for hydrogen evolution [48,63,67,124,125,126,127,128,129,130,131,132,133,134,135,136,137,138,139,140,141,142,143,144].

For the M/S nanocomposite photocatalysts listed in Table 3, their properties for photocatalytic hydrogen evolution achieved remarkable enhancement from the charge separation at the metal-semiconductor interface and the SPR of the metal component. In the work of Bi et al., they utilized Pt-Ni alloy nanoparticles to decorate g-C_3_N_4_ for saving the usage of Pt [124]. The results revealed the PtNi_x_/g-C_3_N_4_ photocatalysts could exhibit comparable properties to that of Pt/g-C_3_N_4_ photocatalysts, which provided a possible approach for developing M/S nanocomposite photocatalysts with excellent properties as well as low cost. In the work of Ingram et al., the difference in the properties of Ag/TiO_2_ and Au/TiO_2_ (TiO_2_ was doped by nitrogen atoms, i.e., N-TiO_2_) photocatalysts for visible light-driven hydrogen evolution was investigated, which shed light on more effective utilization of SPR [125]. As revealed in Figure 10a, the SPR of Ag and Au was excited at the wavelengths of about 400 and 500 nm, respectively, while the absorption edge of N-TiO_2_ was around 400 nm. Therefore, the SPR of Ag was more efficient for exciting electron-hole pairs in N-TiO_2_ than Au, which was supported by the photocatalytic hydrogen evolution test results in Figure 10b.

Towards the core-shell and yolk-shell structure, Ma et al. investigated the effect of Au cores on the properties of core-shell Au@CdS photocatalysts for hydrogen evolution [136]. As revealed in Figure 11, under irradiation of visible light with wavelength ≥ 420 nm, core-shell Au@CdS exhibited apparently higher photocatalytic properties than pristine CdS, while under irradiation of visible light with wavelength ≥500 nm, the photocatalytic properties of core-shell Au@CdS were only slightly higher than pristine CdS. Thus, Ma et al. thought the radiative energy transfer from SPR-excited Au to CdS was the main contribution to the enhanced photocatalytic properties of core-shell Au@CdS, rather than the electron transfer from SPR-excited Au to CdS. In the work of Ngaw et al., the effect of Au content on the photocatalytic properties of yolk-shell Au@TiO_2_ for water splitting was investigated [137]. The photocatalytic properties of yolk-shell Au@TiO_2_ also did not improve linearly with the Au content under the irradiation of both visible light and UV light, and reached a peak at 2 wt% Au content. In addition, Ngaw et al. ascribed the enhanced photocatalytic properties of yolk-shell Au@TiO_2_ to the void space and highly porous shell in yolk-shell Au@TiO_2_, which provided more active sites for H^+^ ions to be reduced and more channels for reactants to diffuse into and out of the photocatalytic particles.

In the work of Seh et al., the properties of Janus Au/TiO_2_ and core-shell Au@TiO_2_ photocatalysts for hydrogen evolution were compared and investigated [63]. As revealed in Figure 12a,b, Janus Au/TiO_2_ photocatalysts exhibited higher properties than core-shell Au@TiO_2_ photocatalysts and the properties of Janus Au/TiO_2_ photocatalysts increased with the size of Au nanoparticles. Moreover, Seh et al. attributed the enhanced charge separation in Janus Au/TiO_2_ photocatalysts to the strong plasmonic near-fields localized closely to the Au-TiO_2_ interface, which was supported by the discrete-dipole approximation simulation results.

As for the array structure, Wang et al. demonstrated the excellent properties of multi-segmented Au/CdS nanorod arrays (NRAs) for photocatalytic hydrogen evolution [67]. As revealed in Figure 13a, the activities of Au/CdS NRAs for hydrogen evolution increased with the number of Au-CdS segments under irradiation of simulated sunlight. In addition, compared with pristine CdS NRAs, the activities of Au/CdS NRAs for hydrogen evolution gained significant enhancement under irradiation of simulated sunlight, which was presented in Figure 13b. Wang et al. attributed the excellent properties of Au/CdS NRAs for hydrogen evolution to the charge separation at the Au-CdS interface under irradiation of simulated sunlight.

Toward the multi-junction structure, Yu et al. demonstrated the excellent properties of multi-junction CdS/Au/ZnO photocatalysts for hydrogen evolution [139]. To reveal the high charge separation efficiency bought by the Z-scheme electron transfer, Yu el al. also synthesized Au/CdS/ZnO photocatalysts by depositing Au on CdS/ZnO photocatalysts. As shown in Figure 14a, the properties of multi-junction CdS/Au/ZnO photocatalysts was 4.5 times higher than that of CdS/ZnO photocatalysts, while the properties of Au/CdS/ZnO photocatalysts achieved only a small enhancement compared with CdS/ZnO photocatalysts. The PL spectra in Figure 14b further confirmed the high charge separation efficiency due to the Z-scheme electron transfer in multi-junction CdS/Au/ZnO photocatalysts.

### 4.4. Disinfection

In 1985, Matsunaga et al. first reported the photocatalytic inactivation of bacteria on the surface of TiO_2_ [145]. Since then, a lot of research has been devoted to the photocatalytic inactivation of microorganisms such as bacteria, viruses, protozoa and so on [18,146,147,148,149,150]. Furthermore, due to the nonexistence of secondary pollution, photocatalytic disinfection is a promising alternative approach for water purification. Table 4 lists some of the most representative M/S nanocomposite photocatalysts for disinfection [151,152,153,154,155,156,157,158,159].

Taking into account the practical application of photocatalytic disinfection for water purification, most of the light sources involved in the research are visible light as revealed in Table 4. Therefore, the enhanced properties of M/S nanocomposite photocatalysts for disinfection are mainly attributed to the SPR of the metal component. In the work of Shi et al., due to the SPR of Ag, the Ag/AgX (X = Cl, Br, I) photocatalysts exhibited remarkable properties for the inactivation of *Escherichia coli* under visible light illumination [157]. Shi el al. also evaluated the contribution of different photo-generated reactive species to the disinfection process by adding scavengers into the reaction system. The results indicated holes are the dominant reactive species over other reactive species such as electrons, ^•^OH, H_2_O_2_ and so on.

## 5. Conclusions and Perspectives

In this review, we demonstrate the properties of M/S nanocomposite photocatalysts in relation to their structures for application in environmental remediation, selective organic transformation, hydrogen evolution and disinfection. Due to the enhanced charge separation at the metal-semiconductor interface and increased absorption of visible light induced by the SPR of metals, M/S nanocomposite photocatalysts usually exhibit much better properties than pristine semiconductor photocatalysts.

For future development of M/S nanocomposite photocatalysts, our perspectives can be summarized as the following four points:(1)To date, most of metals utilized to combine with semiconductor photocatalysts are noble metals, which are scarce in nature and expensive. To save the use of noble metals, coupling semiconductors with alloys composed of noble and non-noble metals is highly recommended. In addition, due to the non-linear relationship between the properties of M/S nanocomposite photocatalysts and metal loading, precise control over the metal loading in the M/S nanocomposite photocatalysts deserve further research.(2)For more efficient utilization of SPR to enhance the properties of M/S nanocomposite photocatalysts, the SPR excitation wavelength of the metal nanoparticles should overlap the absorption edge of the semiconductor nanoparticles, which could be achieved by changing the shape and particle size of the metal nanoparticles and modulating the band structure of the semiconductor nanoparticles.(3)Synergistic utilization of enhanced charge separation at the metal-semiconductor interface and SPR of metals might endow the M/S nanocomposite photocatalysts with even better properties under visible light illumination, because the SPR-induced charge separation in M/S nanocomposite photocatalysts could be further enhanced by introducing another metal co-catalyst with large work function to the M/S nanocomposite photocatalysts.(4)Due to their intrinsic ability to prohibit particle agglomeration, the core-shell, yolk-shell and array structures (especially the array structure) might be the ideal structures for M/S nanocomposite photocatalysts. Therefore, there exists a strong demand for more facile synthesis of these structures.

## Figures and Tables

**Figure 1 nanomaterials-09-00359-f001:**
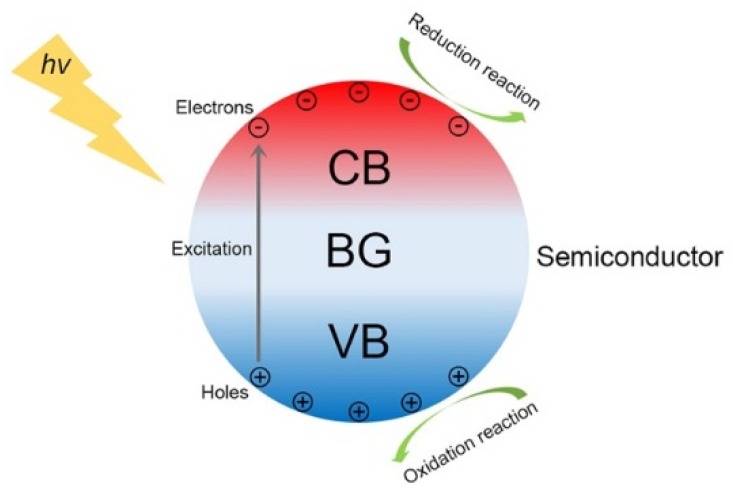
Schematic diagram for the photocatalytic process.

**Figure 2 nanomaterials-09-00359-f002:**
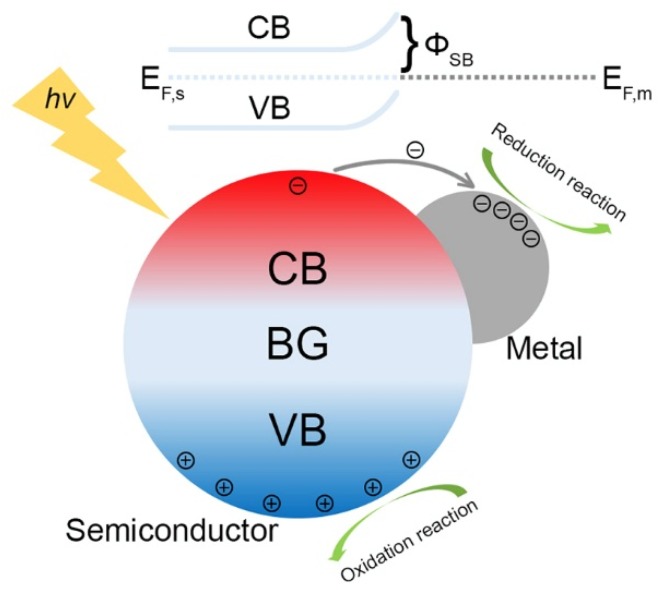
Enhanced charge separation at the metal-semiconductor interface.

**Figure 3 nanomaterials-09-00359-f003:**
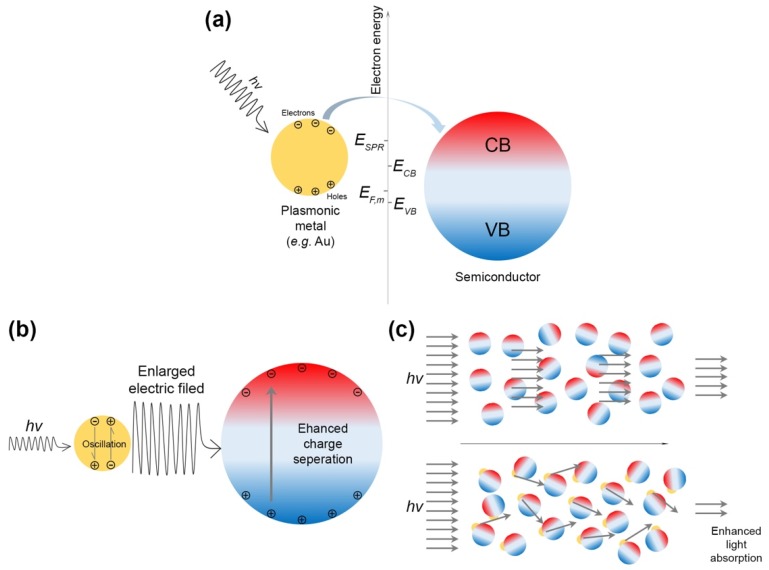
(**a**) SPR-induced electron injection; (**b**) Charge separation induced by NFEF; (**c**) Scattering-enhanced light absorption.

**Figure 4 nanomaterials-09-00359-f004:**
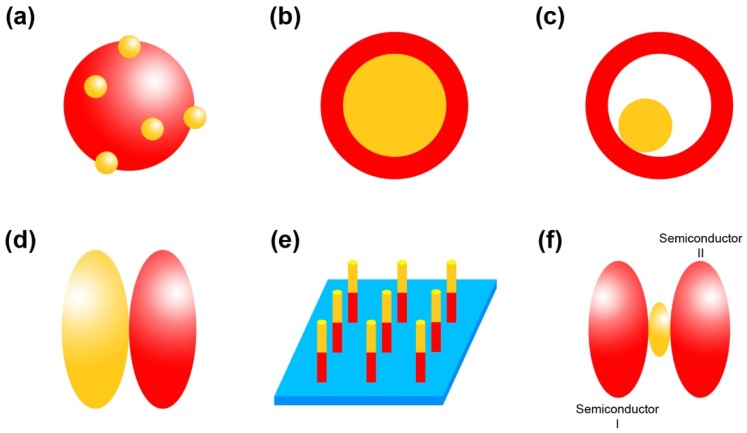
(**a**) Conventional structure; (**b**) Core-shell structure; (**c**) Yolk-shell structure; (**d**) Janus structure; (**e**) Array structure; (**f**) Multi-junction structure.

**Figure 5 nanomaterials-09-00359-f005:**
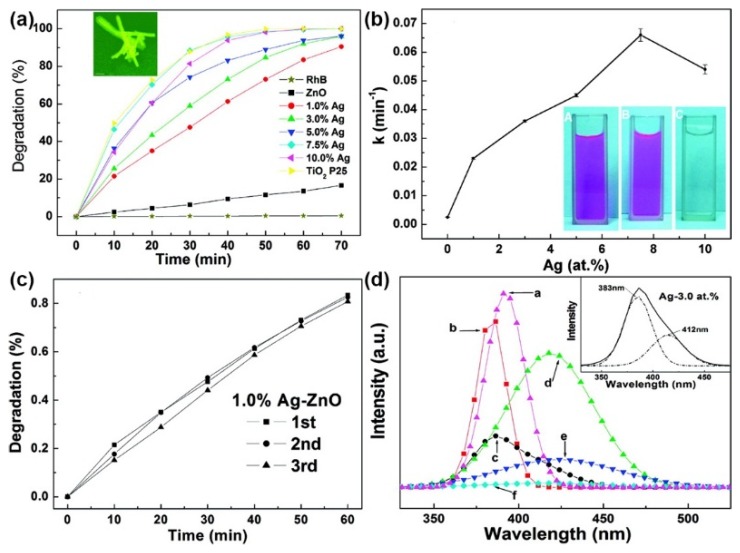
(**a**) Kinetics of the photodegradation of RhB by Ag/ZnO nanoparticles; (**b**) The degradation rate constant versus the Ag loading content; (**c**) Photodegradation of RhB by Ag/ZnO nanoparticles for three cycles; (**d**) PL spectra of the Ag/ZnO nanofibers with various Ag loading contents ((a) pure ZnO, (b) 1 atom% Ag, (c) 3 atom% Ag, (d) 5 atom% Ag, (e) 7.5 atom% Ag, (f) 10 atom% Ag). Reproduced from [79], with copyright permission from American Chemical Society, 2009.

**Figure 6 nanomaterials-09-00359-f006:**
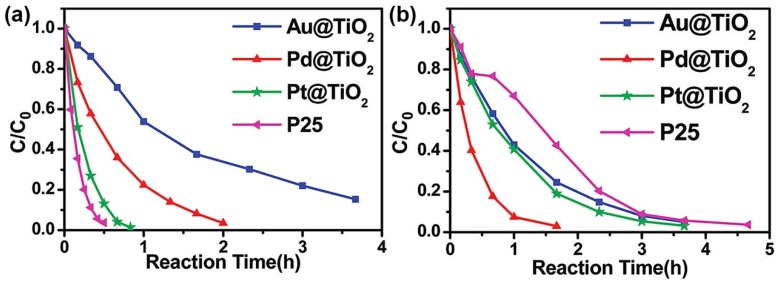
(**a**) Degradation of RhB by P25 and M@TiO_2_ (M = Au, Pd and Pt) under UV irradiation; (**b**) Degradation of RhB by P25 and M@TiO_2_ (M = Au, Pd and Pt) under visible light irradiation. Reproduced from [90], with copyright permission from American Chemical Society, 2011.

**Figure 7 nanomaterials-09-00359-f007:**
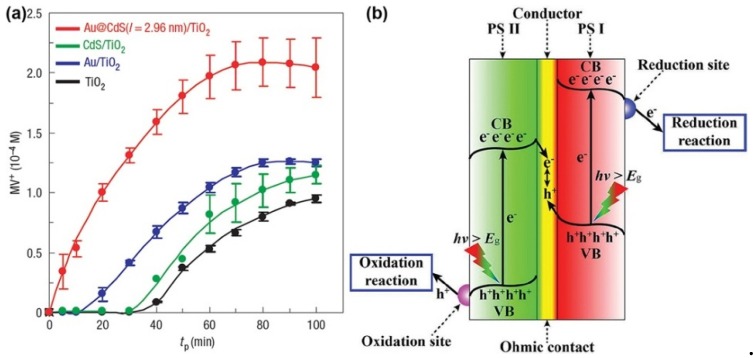
(**a**) Photocatalytic reduction of MV by TiO_2_, Au/TiO_2_, CdS/TiO_2_ and Au@CdS/TiO_2_ (CdS/Au/TiO_2_). Reproduced from [71], with copyright permission from Springer Nature, 2006; (**b**) Schematic diagram of Z-scheme electron transfer. Reproduced from [68], with copyright permission from John Wiley & Sons, 2014.

**Figure 8 nanomaterials-09-00359-f008:**
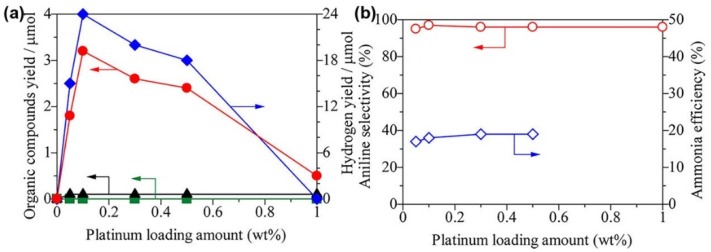
(**a**) Conversion yield of Pt/TiO_2_ versus the Pt loading amount; (**b**) Aniline selectivity of Pt/TiO_2_ versus the Pt loading amount. Reproduced from [107], with copyright permission from American Chemical Society, 2013.

**Figure 9 nanomaterials-09-00359-f009:**
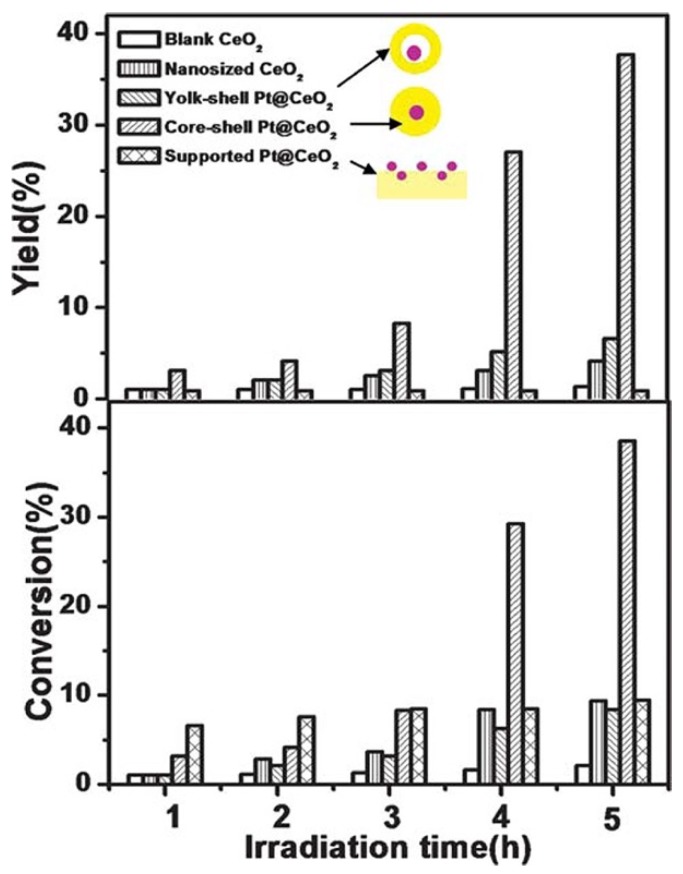
Results of photocatalytic selective oxidation of benzyl alcohol to benzyl aldehyde over the core-shell Pt@CeO_2_, yolk-shell Pt@CeO_2_, supported Pt/CeO_2_, CeO_2_ nanoparticles and blank CeO_2_. Reproduced from [110], with copyright permission from The Royal Society of Chemistry, 2011.

**Figure 10 nanomaterials-09-00359-f010:**
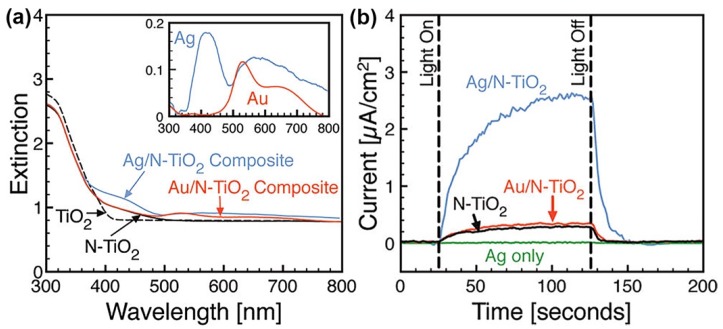
(**a**) Ultraviolet-visible extinction spectra of TiO_2_, N-TiO_2_, Ag/N-TiO_2_ and Au/N-TiO_2_ samples. The inset shows difference spectra for Ag and Au (i.e., the Ag/N-TiO_2_ or Au/N-TiO_2_ spectrum minus the N-TiO_2_ spectrum); (**b**) Photocurrent responses (per macroscopic electrode area) under visible light illumination. Reproduced from [124], with copyright permission from American Chemical Society, 2011.

**Figure 11 nanomaterials-09-00359-f011:**
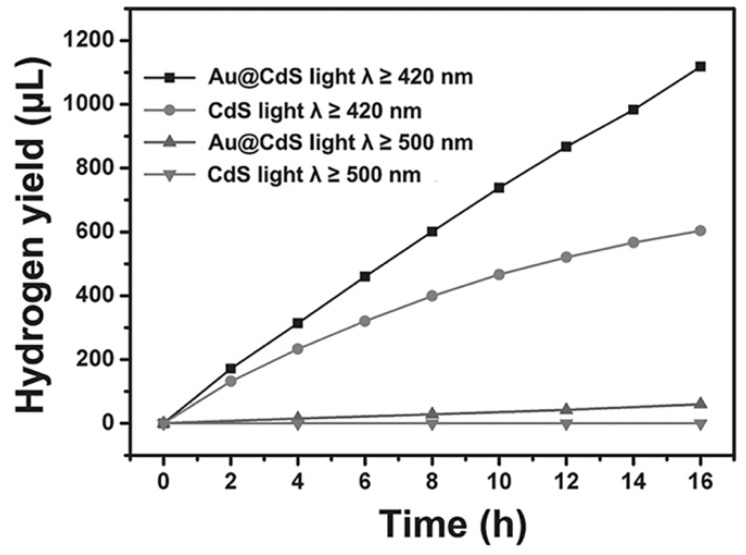
Photocatalytic water splitting of core-shell Au@CdS and CdS nanoparticles under irradiation of visible light with wavelength more than 420 nm and 500 nm, respectively. Reproduced from [133], with copyright permission from John Wiley & Sons, 2014.

**Figure 12 nanomaterials-09-00359-f012:**
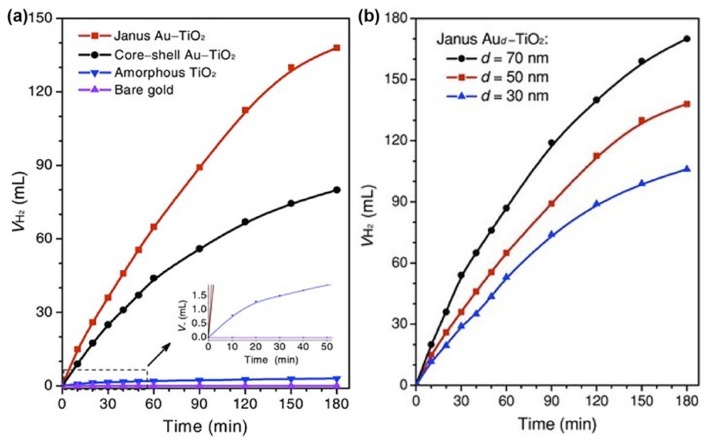
(**a**) Hydrogen evolution through photocatalytic water splitting on Janus Au/TiO_2_, core-shell Au@TiO_2_, amorphous TiO_2_ and pristine Au nanoparticles; (**b**) Hydrogen evolution through photocatalytic water splitting on Janus Au/TiO_2_ with different Au nanoparticle sizes. Reproduced from [63], with copyright permission from John Wiley & Sons, 2012.

**Figure 13 nanomaterials-09-00359-f013:**
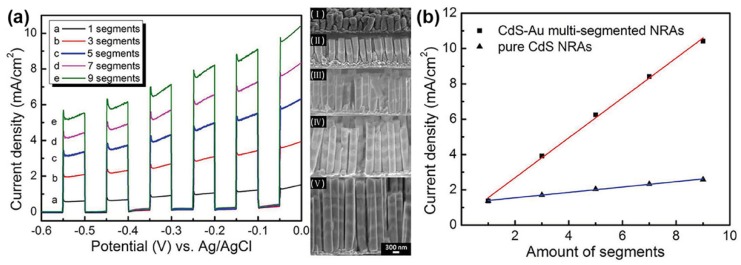
(**a**) Hydrogen evolution on the multi-segmented CdS/Au NRAs in the photoelectrochemical cell under irradiation of simulated sun light. (I-V) SEM images of the side view of the CdS-Au NRAs. (**b**) Comparison of the activities for hydrogen evolution on the CdS/Au NRAs and pristine CdS NRAs under irradiation of simulated sunlight. Reproduced from [67], with copyright permission from John Wiley & Sons, 2014.

**Figure 14 nanomaterials-09-00359-f014:**
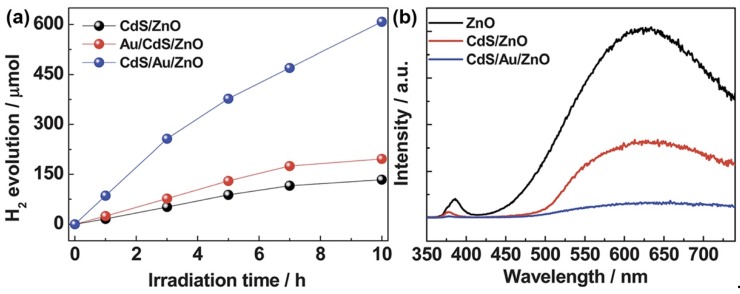
(**a**) Hydrogen evolution through photocatalytic water splitting on CdS/ZnO, Au/CdS/ZnO and CdS/Au/ZnO, respectively. (**b**) PL spectra of ZnO, CdS/ZnO and CdS/Au/ZnO at a 270 nm excitation wavelength. Reproduced from [136], with copyright permission from The Royal Society of Chemistry, 2013.

**Table 1 nanomaterials-09-00359-t001:** Metal/semiconductor (M/S) nanocomposite photocatalysts for environmental remediation.

Photocatalyst	Structure	Pollutant	Light source	Reference
Au/TiO_2_	Conventional structure	RhB	Visible light	[51]
Au/ZnO	Conventional structure	RhB, phenol red, procion red	Ultraviolet (UV) light	[76]
Au/ZnO	Conventional structure	RhB	UV light	[77]
Ag/ZnO	Conventional structure	MB	UV light	[78]
Ag/ZnO	Conventional structure	RhB	UV light	[79]
Ag/ZnO	Conventional structure	RhB	Simulated sunlight	[80]
Au/TiO_2_	Conventional structure	MB, MO, *p*-nitrophenol	Visible light	[81]
Ag/ZnO	Conventional structure	RhB	Visible light	[82]
Au/TiO_2_, Pt/TiO_2_, Ir/TiO_2_	Conventional structure	Azo dye	UV light	[83]
Ag/TiO_2_	Conventional structure	RhB, ciprofloxacin	Visible light	[84]
Cu/Cu_2_O	Conventional structure	RhB, MO, MB	Visible light	[85]
Bi/amorphous bismuth oxide (A-BO)	Conventional structure	NO	Visible light	[86]
Bi/(BiO)_2_CO_3_	Conventional structure	NO	Visible light	[87]
Bi/g-C_3_N_4_	Conventional structure	NO	Visible light	[88]
Bi/ZnWO_4_	Conventional structure	NO	Visible light	[89]
Bi/BiOCl	Conventional structure	NO	Visible light	[90]
M@TiO_2_ (M = Au, Pd, Pt)	Core–shell structure	RhB	UV light, visible light	[91]
Au@Cu_2_O	Core–shell structure	MO	Visible light	[92]
Au@Cu_2_O	Core–shell structure	MO	Visible light	[93]
Au@CdS	Core–shell structure	RhB	Visible light	[94]
Au@TiO_2_	Core–shell structure	Acetaldehyde	UV light, visible light	[95]
Bi@Bi_2_O_3_	Core–shell structure	NO	Visible light	[96]
Ag@Cu_2_O	Core–shell structure	MO	Visible light	[97]
Au@TiO_2_	Yolk–shell structure	RhB	Visible light, simulated sunlight	[59]
Cu@CuS	Yolk–shell structure	MB	Simulated sunlight	[98]
Au/ZnO	Janus structure	RhB	UV light	[99]
Au/ZnO	Janus structure	MO	UV light	[100]
CdS/Au/TiO_2_	Multi-junction structure	Methyl viologen (MV)	UV light	[71]

**Table 2 nanomaterials-09-00359-t002:** M/S nanocomposite photocatalysts for selective organic transformation.

Photocatalyst	Structure	Reaction Type	Reactant	Light Source	Reference
Pt/TiO_2_, Pd/TiO_2_	Conventional structure	Oxidation reaction	Benzene	UV light	[106]
Au/TiO_2_	Conventional structure	Oxidation reaction	Benzene	Simulated sunlight	[107]
Pt/TiO_2_	Conventional structure	Oxidation reaction	Benzene	UV light	[108]
Pt/TiO_2_	Conventional structure	Oxidation reaction	(substituted) Benzene	UV light	[109]
Au/CeO_2_	Conventional structure	Oxidation reaction	Aromatic alcohols	Visible light	[110]
Pt@CeO_2_	Core-shell, yolk-shell structure	Oxidation reaction	Benzyl alcohol	Visible light	[111]
Au/TiO_2_	Janus structure	Oxidation reaction	Methanol	UV light	[112]
Ag/TiO_2_	Conventional structure	Reduction reaction	Nitrobenzene	UV light	[113]
M/TiO_2_ (M = Pt, Pd, Rh, Ag and Au)	Conventional structure	Reduction reaction	Carbon dioxide (CO_2_)	UV light	[114]
Pt-Cu/TiO_2_	Conventional structure	Reduction reaction	CO_2_	Visible light	[115]
Ag/TiO_2_	Conventional structure	Reduction reaction	CO_2_	Simulated sunlight	[116]
Ag/La_2_Ti_2_O_7_	Conventional structure	Reduction reaction	CO_2_	UV light	[117]
Pt/TiO_2_	Conventional structure	Reduction reaction	CO_2_	Simulated sunlight	[118]
Au-Pd/ZrO_2_	Conventional structure	Oxidation, coupling reaction	Benzylamine, benzyl alcohol etc.	Visible light	[119]
Pd/SiC	Conventional structure	Coupling reaction	Iodobenzene, phenylboronic acid	Visible light	[120]

**Table 3 nanomaterials-09-00359-t003:** M/S nanocomposite photocatalysts for hydrogen evolution.

Photocatalyst	Structure	Light Souce	Reference
Pt/TiO_2_	Conventional structure	UV light	[27]
PtNi_x_/g-C_3_N_4_	Conventional structure	UV light	[124]
Au/TiO_2_, Ag/TiO_2_	Conventional structure	Visible light	[125]
Cu/TiO_2_, Ni/TiO_2_, CuNi/TiO_2_	Conventional structure	UV light	[126]
Pt_3_Co/CdS, Pt_3_Co/ TiO_2_	Conventional structure	UV light	[127]
Au/ZnO	Conventional structure	Simulated sunlight	[128]
Au/MoS_2_	Conventional structure	Visible light	[129]
Au/ZnO	Conventional structure	Visible light	[130]
Au/CdS	Conventional structure	Simulated sunlight	[131]
Ni/CdS	Conventional structure	UV light	[132]
Pt/TiO_2_, Au/TiO_2_	Conventional structure	Simulated sunlight	[133]
SnRu_x_/TiO_2_	Conventional structure	UV light	[134]
Au/Cu_2_O	Conventional, core-shell structure	Visible light	[135]
Au@CdS	Core-shell structure	Visible light	[136]
Au@TiO_2_	Yolk-shell structure	UV light	[137]
Au@TiO_2_	Yolk-shell structure	UV light	[138]
Au/TiO_2_	Janus structure	Visible light	[63]
Au/CdS	Array structure	Visible light	[67]
CdS/Au/ZnO	Multi-junction structure	UV light	[139]
CdS/Au/MoS_2_	Multi-junction structure	Visible light	[140]
CdS/Pt/ZnO	Multi-junction structure	Simulated sunlight	[141]
CdS/M/TiO_2_ (M = Au, Ag, Pt, Pd)	Multi-junction structure	Simulated sunlight	[142]
CoO_x_/Ir/Ta_3_N_5_	Multi-junction structure	Simulated sunlight	[143]
Cr_2_O_3_/Rh/ (Ga_1-x_Zn_x_)(N_1-x_O_x_)	Multi-junction structure	Visible light	[144]

**Table 4 nanomaterials-09-00359-t004:** M/S nanocomposite photocatalysts for disinfection.

Photocatalyst	Structure	Bacteria	Light Source	Reference
Ag/g-C_3_N_4_	Conventional structure	*Escherichia coli*	UV and visible light	[151]
Ag/TiO_2_	Conventional structure	*Escherichia coli*	UV light	[152]
Ag/TiO_2_	Conventional structure	*Escherichia coli*	Sunlight	[153]
Ag/TiO_2_	Conventional structure	*Escherichia coli*	Simulated sunlight	[154]
Cu/TiO_2_	Conventional structure	*Escherichia coli*	UV and visible light	[155]
Ag/BiOI	Conventional structure	*Escherichia coli*	Visible light	[156]
Ag/AgX (X = Cl, Br, I)	Conventional structure	*Escherichia coli*	Visible light	[157]
Ag/ZnO	Conventional structure	*Escherichia coli*	Visible light	[158]
Ag@ZnO	Core–shell structure	*Vibrio cholerae*	Sunlight	[159]

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
