# Peer review of "Metal/Semiconductor Nanocomposites for Photocatalysis: Fundamentals, Structures, Applications and Properties"

_nanomaterials, 2019, doi:10.3390/nano9030359_

Round 1

Reviewer 1 Report

The re-submitted version of the manuscript includes new information, higher discusión level that make possible the acceptance in nanomaterial. I would recommend, however, include more references to non-noble metals containing samples such as Cu, Ni, etc. (i.e. Angew.Chem. Int.Ed. 2018, 57,1199 –1203).

Author Response

Thanks very much for your recognition of our resubmitted manuscript, and your recommendation is also very pertinent. We added four references into our review, i.e. Dalton Transactions 2015, 44, 15645-15652;  Angewandte Chemie-International Edition 2018, 57, 1199-1203; ACS Catalysis 2015, 5, 6615-6623; Applied Catalysis B: Environmental 2016, 182, 277-285.

Reviewer 2 Report

Dear authors, for me the the manuscript is fine. It is not an outstangiong contribution. Probalbly it is not needed any more any review like this because there are several in the literature and there are no novelties in the perspectives given by the authors. However, the manuscrit has no further problem. In my view, it can be published in the present form.

Author Response

Thanks very much for your kind comments and suggestions about our resubmitted manuscript.

This manuscript is a resubmission of an earlier submission. The following is a list of the peer review reports and author responses from that submission.

Round 1

Reviewer 1 Report

The conception of the manuscript could be acceptable however the overage is extremely poor and superficial. It has been devoted mainly to very old ideas. There is no novelty or interest in reading this manuscript. There is no interesting focusing ideas..even the literature is old. The environmental aspects cannot be devoted to rhodamine B or methyl orange degradation..Dyes bleaching cannot be considered indeed an environmental application. These are only "easy model substrates" to test some kind of novel materials...In my view the manuscript should be re-written under new basis trying to focus the main achivements using metal/semiconductor materials in photocatalysis and not just describing some articles by random articles...

Reviewer 2 Report

This contribution aims to provide a compressive revision of metal/semiconductor materials for photo-catalytic applications. The manuscript presents interesting points and could be considered for publication after some revisions. Although seem to be a mini-review instead of a typical review, there are several important ideas that authors have to consider before considering this work suitable for publication in Nanomaterials.

- Photo-handling analysis (including charge mobility schemes) of M/S structures must be analysed in details using literature reports. Authors have commented studies in which photo-luminesce and photo-electrochemical measurements are discussed. However, there are no references to EPR measurements. (Chemical Society Reviews: 10.1039/C8CS00108A, Applied Catalysis B: Environmental 183 (2016) 86–95, J. Catal., 2013, 303, 141–155)

- Please, provide more information about bimetallic systems and alloying metallic phase. Applied Catalysis B: Environmental 236, pp. 88-98, Journal of Molecular Catalysis 2018 10.1016/j.mcat.2018.11.011. See, for example, contributions of Prof. Fernández-García and Prof. Fornasiero.

- Beside “environmental remediation, selective organic transformation and water splitting for hydrogen evolution” M/S composite materials have shown outstanding photo-disinfection capability. I would strongly recommend include some works for this application. See contributions of Prof. Fernández-García and Prof. Pulgarín.

- g-C3N4-related materials have shown outstanding results in all photo-catalytic applications featured in this review. Photo-handling behaviour of metal/g-C3N4 materials follows different pathway in comparison with traditional photocatalysts such as TiO2 and ZnO. I would recommend authors include references of these kinds of structures.

- In addition to above-commented references, this reviewer recommends taking into account relevant contributions about M/S nanocomposites reported by others groups considered experts in photocatalysis: Prof. Fernández-García, Prof. Fornasiero, Prof. Keller, Prof. Rose Amal and others.

Reviewer 3 Report

The proposed Review article titled "Metal/Semiconductor Nanocomposites for Photocatalysis: Fundamentals, Structures, Applications and Properties" gives a clear presentation of Metal/Semiconductor photocatalysts design and their working principles. Also presents the improved photocatalytic activity after metals loading. Actually there were many reviews on the Metal/semiconductor nanostructures and their principles behind the working as a photocatalysts.

For example,

https://pubs.acs.org/doi/abs/10.1021/jacs.6b10288

https://pubs.rsc.org/en/content/articlelanding/2018/nr/c7nr08487k#!divAbstract

The current review lacks several scientific aspects such as 

why the need of metal loading for semiconductors, structural and morphological modifications after loading, how the metal loading will be benefitted...etc. 

Introduction lacks motive of review article and why the metal loading necessary for photocatalyst field.

Extensive citations will be needed, for example how the metal particle will be useful for improving the activity. Authors should chek the previously published results,

For example.

https://pubs.acs.org/doi/abs/10.1021/ja200086g

https://pubs.acs.org/doi/abs/10.1021/jp011118k

https://pubs.acs.org/doi/abs/10.1021/ja305603t

https://pubs.acs.org/doi/10.1021/acssuschemeng.8b00249

https://pubs.acs.org/doi/10.1021/cm802074j

https://www.nature.com/articles/nmat1734

https://onlinelibrary.wiley.com/doi/pdf/10.1002/adfm.200305430

.